# Relationship between Dynamic Balance and Physical Characteristics and Functions in Elite Lifesaving Athletes

**DOI:** 10.3390/jfmk9030134

**Published:** 2024-08-08

**Authors:** Shota Ichikawa, Tsukasa Kumai, Yui Akiyama, Takumi Okunuki, Toshihiro Maemichi, Masatomo Matsumoto, Zijian Liu, Ryusei Yamaguchi, Hiroyuki Mitsui, Kai Suzuki, Hisateru Niki

**Affiliations:** 1Graduate School of Sport Sciences, Waseda University, 2-579-15 Mikajima, Tokorozawa 359-1192, Saitama, Japan; 2Department of Orthopaedic Surgery, St. Marianna University School of Medicine, 2-16-1 Sugao, Miyamae, Kawasaki 216-8111, Kanagawa, Japan; 3Faculty of Sport Sciences, Waseda University, 2-579-15 Mikajima, Tokorozawa 359-1192, Saitama, Japan

**Keywords:** sand training, sandy beach, Y-balance test, lifesaver

## Abstract

Balance is important in lifesaving competitions. We aimed to investigate the relationship between dynamic balance and physical characteristics and functions in elite lifesavers by measuring the foot muscle cross-sectional area, ankle joint muscle strength, toe grasp strength, plantar superficial sensation, and dynamic balance (Y-balance test). In this observational study, we measured the foot muscle cross-sectional area, ankle dorsiflexion, plantar flexion, external flexion, isometric muscle strength, toe grasp strength, and superficial foot sensation of 15 adult lifesavers (12 males). The results show that toe grasp strength and ankle plantar flexion isometric muscle strength are particularly important for the dynamic balance of elite lifesavers working on sandy surfaces. Sand training improves intrinsic muscle strength and dynamic balance function. However, high training intensity may reduce plantar surface sensation; this needs to be verified through sand training interventions in the future.

## 1. Introduction

Lifesaving is a sport in which participants compete in activities at sea, such as rescue and basic life support. These events are certified by the International Life Saving Federation and are divided into surf events, performed at the seaside, and beach events, performed on the beach [1]. Lifesaving is recognized by the International Olympic Committee but is not yet included in the Olympic program.

Lifesaving competitions are performed barefoot in environments with unstable footing, such as beaches. These environments have attracted attention because they increase physical strength, leg strength, balance function, proprioceptive sensation, and spatial cognitive ability [2,3,4]. However, few studies have been conducted on lifesaving.

Compared with non-competing adults, lifesavers have more developed intrinsic muscles in their feet [5], suggesting that the cross-sectional area and strength of the toe grasp may be important for sand sports. Lifesavers run on unstable footing and may require a higher balance function compared to non-competitors, using complex factors such as toe grasp strength, muscle strength around the ankle joint, ankle joint position sense, and plantar superficial sensation during foot contact on sandy ground. However, it is not certain which foot functions influence the dynamic balance of athletes in athletic environments with unstable footing compared with athletes in other sports on hard ground [3,6,7,8]. We, therefore, aimed to clarify the physical characteristics of lifesavers operating in an unstable environment, as well as the effects of sand training, by evaluating foot morphology and function. Our hypothesis posits that the intrinsic muscles of the foot play a crucial role in enhancing dynamic balance through improved toe grasp strength and ankle stability. These adaptations may be pivotal for athletes performing on uneven sandy terrain, distinct from those required on stable, hard surfaces. By integrating the concept of ‘bilateral outcomes of unilateral injuries’, as suggested by recent studies [9,10], we further explore how injuries might affect proprioceptive adaptations across both affected and unaffected limbs, potentially altering balance control strategies even in the healthy limb. This holistic approach to studying balance and foot functionality in lifesavers could provide invaluable insights into injury prevention and performance optimization in sand-based sports.

## 2. Materials and Methods

### 2.1. Participants and Design

Fifteen elite lifesavers (12 males and 3 females, ≥18 years old) belonging to a lifesaving club were included. These lifesaver athletes are members of strong teams that regularly compete in the All-Japan Championships. Two participants have been practicing lifesaving for 2 years, three for 3 years, four for 4 years, and one participant each for 6, 8, 11, 14, 15, and 17 years. At the time that they were assessed, the volume of training was five sessions per week and three hours per week. We excluded lifesavers with a history of lower limb surgery; lower limb trauma or disability experienced within the past 6 months; and those with diabetes, joint diseases, or neurological diseases. The foot muscle cross-sectional area, ankle joint muscle strength, toe grasp strength, plantar superficial sensation, and muscle activity were measured. Fifteen participants were required for the present study design to achieve a power of 80%, an effect size (r) of 0.7, and an alpha level of 0.05.

The study was approved by the Ethics Committee of Waseda University (no. 2020-227). The participants were fully informed about the experiment before it occurred, and their written consent was obtained.

### 2.2. Ultrasound Imaging

The muscle cross-sectional area (short-axis image) was acquired in B-mode using an ultrasound imaging system (Aplio a Verifia, Canon Medical Systems Corporation, Tokyo, Japan) and a linear probe (L55, 7.5 MHz). The participants were instructed to sit and place both feet on a plate filled with water in a tank made of polymethylpentene resin, with the ankle joint in 0° dorsiflexion and the knee joint in 90° flexion. The weight of the upper body was placed behind the hands to minimize the application of weight on the foot [5,11,12]. The intrinsic (abductor hallucis brevis, abductor digiti minimi, flexor digitorum brevis, and flexor hallucis brevis) and extrinsic (peroneal longus and brevis, flexor hallucis longus, tibialis anterior, extensor digitorum longus, tibialis posterior, and flexor digitorum longus) muscles of the foot were measured. The extrinsic muscles of the foot and the abductor digiti minimi were measured by applying the ultrasound probe to the skin, while the intrinsic muscles of the foot were measured from the plantar surface of the tank. The pressure from the probe was kept to a minimum to prevent the exertion of force on the tissues, and ultrasound gel was applied to the skin and probe for all measurements. The probe was removed for each measurement of each muscle, and three measurements were obtained, with the average value used for analysis. The ultrasound scanning protocol was performed in accordance with previous studies’ protocols [5].

### 2.3. Ankle Isometric Muscle Strength Testing

Isometric muscle strength during dorsiflexion, plantar flexion, and eversion was measured using a handheld dynamometer (HHD) (ANIMA, Tokyo, Japan). For dorsiflexion muscle strength, the participant was instructed to lie supine, with their hip joint in 0° internal/external rotation, and to grasp the sides of the bed. The belt attached to the HHD was fixed to the dorsal metatarsal head at 0° plantar dorsiflexion of the ankle joint, and the position was adjusted such that the belt and the direction of muscle exertion were parallel. For plantar flexion muscle strength, the participant was instructed to kneel with their knee joint at 90° and place their hands on the ground. The belt attached to the HHD was fixed to the plantar side of the metatarsal head at 0° plantar dorsiflexion of the ankle joint, and the position was adjusted such that the belt and the direction of muscle exertion were parallel. To measure eversion strength, a pillow was placed under the lower leg, such that the right hip joint was in 0° of abduction in the left lateral recumbent position. The HHD was grasped and fixed to the base of the fifth metatarsal. Measurements were performed three times for a maximum muscle exertion period of 5 s. The mean value, normalized to body weight, was used as the representative value [13].

### 2.4. Toe Grasp Force Measurement

A pressure sensor (BIGMAT, NITTA, Osaka, Japan) was used to measure toe pressure. The right side of the right foot was measured in all cases. The load on both legs was equalized at shoulder width during the measurement, and the participants were instructed to flex their toes toward the heel as much as possible while pressing hard on the ground (Figure 1). The maximum pressure of the big toe and toes 2–5 combined was calculated, and the average value of the five measurements was divided by the body weight, normalized as a percentage of body weight (% BW), and used for the analysis.

### 2.5. Superficial Sensory Measurements

Superficial sensory measurements of the feet were performed using a Semes–Weinstein monofilament (Sakai Medical Co., Ltd., Tokyo, Japan) and were taken at nine plantar points: the basolateral side of the fifth phalanx; the basolateral side of the third phalanx; the basolateral side of the first phalanx; the basolateral side of the fifth metatarsophalangeal (MTP) joint; the third MTP joint; the first MTP joint; the basolateral side of the fifth metatarsal base; the basolateral side of the navicular bone; and the central heel area (Figure 2). The participants closed their eyes and were placed in a supine position with the knee extended and the foot in a supinated position. Nylon filaments of varying diameter were used to apply tactile stimulation to the skin. The examiner lowered the filament vertically from 2.5 cm above the foot’s plantar for 1.5 s and applied pressure until the filament was deflected for 1.5 s, according to protocols used in previous studies [14]. The participants were instructed to respond by raising their voice or hand when they felt filament contact; if they did not respond, the filament thickness was increased. The measurements were performed three times with the same filament at each location, and the value of any filament that gained a response on all three occasions was recorded. A logarithmic representation of the force required to flex and deflect when pressure was applied in the longitudinal direction was assigned to each filament as the filament number (using sets of 20 filaments from 1.65 to 6.65). Filament size was defined as log (10 × force in milligrams).

### 2.6. Dynamic Balance Measurement

The maximum reaching distances to the anterior, posterior lateral, and posterior medial sides were measured using a Y-balance test kit (Perform Better Japan, Cranston, RI, USA) on stable ground, with the right leg supported. Limb position with both hands fixed to the waist was measured three times in each direction. If the outstretched leg touched the ground, the hands left the waist, or the heel of the supporting foot lifted, the measurement was judged to have failed and was repeated.

The value obtained by dividing the sum of the maximum reach distances in the three directions was recorded, normalized by the lower limb length, and used as the representative value in the analysis. Lower limb length was measured as the distance from the superior anterior iliac spine to the medial cuticle using a tape measure [15]. A surface electromyogram was attached during measurement, and muscle activity at maximum reach was assessed. The participants held a maximum reaching position for 1 s, and muscle activity during that 1 s period was recorded.

### 2.7. Surface Muscle Activity Analysis

The activity of the peroneal muscles of the foot was recorded and analyzed during the dynamic balance test using Biosignal splux (PLUX wireless Biosignals, Lisbon, Portugal) and Gelled Self-Adhesive Disposable Ag/AgCl Electrodes (PLUX wireless Biosignals Portugal). The muscles measured were the abductor digiti adductus (AH), abductor digiti minor (AD), tibialis anterior (TA), peroneus longus (PL), peroneus brevis (PB), medial gastrocnemius (MG), and soleus (SL), with electrodes applied at the following locations: AH, one transverse finger below the navicular tubercle; AD, one transverse finger behind the base of the fifth metatarsal; TA, proximal third of the line connecting the lateral tibial condyle to the medial tubercle; PL, proximal quarter of the line connecting the fibular head to the external tubercle; PB, distal quarter of the line connecting the fibular head to the external tubercle; MG, site of maximum bulging at the medial gastrocnemius head when standing on tiptoes; and SL, site of maximum bulging at the medial head of the gastrocnemius muscle (distal third of the line connecting the medial epicondyle of the femur to the medial condyle of the femur) [16]. In the Y-balance test, the average muscle activity measured during 1 s of maximal reach was normalized using the maximum voluntary isometric contraction, and the average of three successful trials was recorded.

### 2.8. Statistical Analyses

IBM SPSS Statistics 26 (IBM Corp., Armonk, NY, USA) was used for statistical analysis. The correlations between the basic information (age, height, weight, and body mass index [BMI]), muscle cross-sectional area, ankle isometric strength, toe grasp strength, plantar superficial sensation, muscle activity, and total reach distance in the three directions of the dynamic balance test were investigated. The Shapiro–Wilk test was used to test for normality. Pearson’s correlation coefficient was used when a normal distribution was assumed for the correlations, and Spearman’s correlation coefficient was used when no normal distribution was assumed. The significance level was set at *p* < 0.05.

## 3. Results

### 3.1. Participant Characteristics

Our participants had a mean age of 24.5 ± 5.4 years, height of 169.0 ± 7.1 cm, weight of 66.1 ± 11.4 kg, and BMI of 23 ± 2.5 kg/m^2^ (Table 1). The recorded values of each measurement are shown in Table 2 and Table 3. The dynamic balance test measurements of the maximum forward, backward–outward, and backward–inward reach distances, normalized by leg length and their correlation with each physical characteristic, are shown in Table 4.

### 3.2. Ultrasound Imaging

There was no significant correlation between the muscle cross-sectional area and reach distance in any direction.

### 3.3. Ankle Isometric Muscle Strength Measurements

There was a significant correlation between plantar flexor strength and the posterior medial reach distance (*p* < 0.05), posterior lateral reach distance (r = 0.588, *p* < 0.05), and total reach distance in all three directions (*p* < 0.05).

### 3.4. Toe Grasp Strength

The anterior reach distance tended to be longer with increased toe grasp strength (*p* = 0.057); the posterior medial reach distance increased with higher toe grasp strength (*p* < 0.05); the posterior lateral reach distance increased with higher toe grasp strength (*p* < 0.05); and the total reach distance in the three directions increased with higher toe grasp strength (*p* < 0.05).

### 3.5. Plantar Superficial Sensation

The anterior reach distance increased with the dominant plantar superficial sensation of the third MTP joint (*p* < 0.05). The posterior medial reach distance increased with the plantar superficial sensation of the first and fifth MTP joints (both *p* < 0.05). The posterolateral reach distance tended to increase with the dominant plantar surface sensation at the first and fifth MTP joints (*p* = 0.051 and *p* = 0.077, respectively). The total reach distance in the three directions was greater than the sensation in the first and fifth MTP joints (both *p* < 0.05).

### 3.6. Muscle Activity

The reach distance in the posterolateral direction became shorter as the muscle activity of the abductor hallucis muscle increased (*p* < 0.05). There was no significant correlation between the reach distance and muscle activity in the other directions.

## 4. Discussion

We examined the values of key physical indices, such as the foot muscle cross-sectional area, ankle isometric muscle strength, toe grasp strength, plantar superficial sensation, and muscle activity, as well as the dynamic balance test (Y-balance test), in lifesavers. The Y-balance test values of the lifesavers were related to toe grasp strength, ankle plantar flexion muscle strength, and plantar superficial sensation.

The Y-balance test values of the lifesavers differed from those reported in a previous study involving 25 participants within the same age group from the general population [17]. Dynamic postural control is stabilized by improving the gluteal, thigh, and trunk muscles [18], and the intrinsic foot muscles provide dynamic stability during the propulsive phase from plantar ground contact to toe-off [19]. Lifesavers have well-developed intrinsic foot muscles [5]. However, in this study, differences in the muscle cross-sectional area did not correlate with dynamic balance. In contrast, ankle plantar flexion and toe grasp strength significantly correlated with Y-balance test values in terms of muscle strength. In the backward-reaching movement, the ankle joint is dorsiflexed, the lower limb moves backward, and the center of gravity remains forward. The center of gravity then shifts backward during the process, but is stabilized by the plantar flexor muscle force supporting the weight shift and the toe grasping force grasping the ground. Plantar flexor muscle strength is correlated with dynamic balance control [20], and similar results have been reported in lifesavers.

The involvement of toe grasp strength in balance function has also attracted attention, and this trait has been trained during rehabilitation. Menz et al. [21] and Yoshimoto et al. [22] examined the relationship between dynamic balance ability and toe flexor muscle strength in community-dwelling older people; a decline in dynamic balance ability was associated with a decline in toe flexor muscle strength. A study of university basketball players also reported that toe grasp strength improved with balance training [23], suggesting that toe grasp strength and plantar flexor muscle strength are important for dynamic balance function.

Regarding plantar superficial sensation, there was a significant correlation between sensitivity at the first, third, and fifth MTP joints and the Y-balance test value. While standing, the only areas in contact with the floor are the plantar surfaces of the feet, where numerous sensory receptors exist for postural regulation [24], though these receptors are distributed in different areas [25]. These plantar mechanoreceptors are the only mechanoreceptors that are in contact with the ground and receive sensory information related to ground reaction forces. This sensory information is important for spatial position perception and significantly contributes to balance function [26].

A good sense of touch under the first metatarsal head was found to correlate with somatosensory perception and dynamic balance function in different parts of the plantar surface [27], with cutaneous sensation and muscle strength being related to static and dynamic balance [20]. We also showed that poor sensation of the plantar surface causes a decrease in balance function, and that the plantar surface sensation on the inside and outside of the plantar surface is involved in postural sway due to the appearance of internal and external postural sway during the Y-balance test.

The loss of sensory input from the plantar surface also affects foot muscular activity [28]. Overall, lifesavers tended to have lower plantar superficial sensation. During sand training, barefoot athletes compete in hot, sandy conditions, and the plantar skin is continuously stimulated, causing it to thicken in response to the environment. As plantar mechanoreceptors are located in the dermis layer of the skin, when the skin thickens, the stratum corneum of the epidermis also thickens, which may reduce the superficial plantar sensation, resulting in reduced balance function due to reduced plantar superficial sensation. However, lifesavers may be able to maintain high balance function by using toe grasp strength and periarticular muscular strength.

According to electromyography results, the maximum reach distance in the posterolateral direction decreased as the activity of the abductor pollicis brevis muscle increased. Muscle activity during the Y-balance test in participants with tendinogenesis was highest in the posterior and lateral directions of the TA muscle, followed by that of the PB tendon. Conversely, activity was low in the anterior and medial directions [29]. In our lifesavers, a negative correlation was found between an increase in the muscle activity of the abductor pollicis brevis at maximum reach and a decrease in the maximum reach distance, suggesting that the abductor pollicis brevis muscle is active when there is postural sway at plantar ground contact during maximal reach, as per a previous study [30]. Lifesavers develop intrinsic foot muscles through sand training and utilize toe grasp and ankle plantar flexor muscle strength to stabilize the plantar surface. They show higher Y-balance test scores compared to others in the same age group, suggesting that prolonged sand training may cause a loss of plantar surface sensation associated with thickening of the plantar skin. This is useful for improving foot muscle strength and dynamic balance function.

The limitations of this study include the fact that no comparison with the general population was made, and we assessed the balance function of only one leg per athlete. In addition, the direct causal relationship between lifesaving activities and improvement in balance function is unclear because the study was only conducted at a single time point, and the evaluation of intrinsic sensation and toe grasp strength was not sufficiently divided into intrinsic and extrinsic muscles. Therefore, additional studies are warranted in the future.

## 5. Conclusions

We examined the relationship between the physical characteristics of lifesavers and their dynamic balance function, to investigate the effects of a competitive environment on foot function on sandy terrain. Toe grasp strength and ankle plantar flexor muscle strength are particularly important for dynamic balance in elite lifesavers operating on sand. Sand training improves intrinsic muscle strength and dynamic balance function; however, high training intensity may cause a decrease in plantar superficial sensation, which needs to be verified with training interventions in the future.

This study significantly contributes to both the enhancement of lifesaving athletes’ performance and the fields of sports medicine and exercise science. By elucidating the methods and effects of balance training on sand, the study provides concrete evidence that can aid in improving sports performance and preventing injuries among athletes. Consequently, this research can lead to the optimization of training programs for athletes involved in lifesaving and other sand-based sports.

## Figures and Tables

**Figure 1 jfmk-09-00134-f001:**
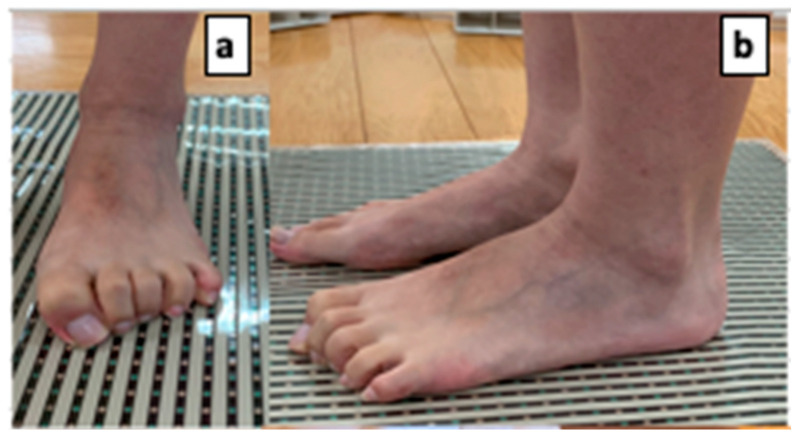
Toe grip strength measurement. (**a**,**b**) show photographs of the legs in which the activity of the leg muscles is dominant. To measure the muscles of the foot, the participants were instructed to bend their toes toward the heel as much as possible while pressing strongly against the ground.

**Figure 2 jfmk-09-00134-f002:**
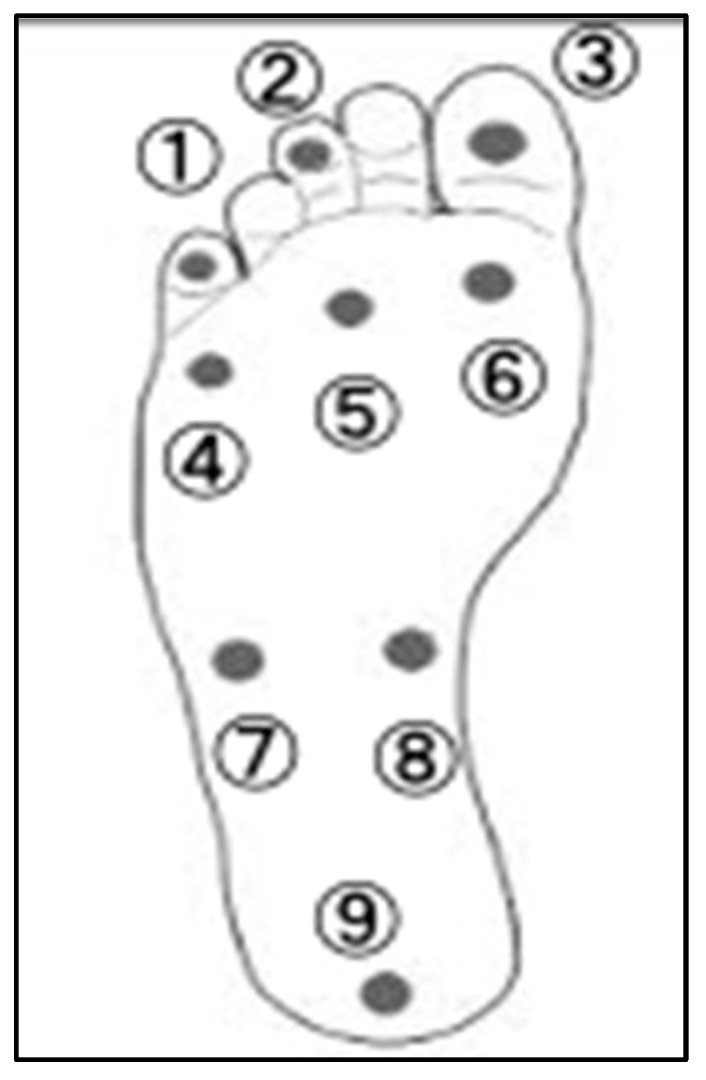
Measurement points for plantar surface perception. (1) Base of 5th toe; (2) base of 3rd toe; (3) base of 1st toe; (4) 5th metatarsal joint; (5) 3rd metatarsal joint; (6) 1st metatarsal joint; (7) base side of 5th metatarsal base; (8) base side of navicular bone; (9) base in center of heel.

**Table 1 jfmk-09-00134-t001:** Participant characteristics.

Variable	Lifesavers
	(N = 15; 12 Males, 3 Females)
	Mean (SD)
Age (years)	24.5 (5.4)
Height (cm)	169 (7.1)
Weight (kg)	66.1 (11.4)
BMI (kg/m^2^)	23 (2.5)

BMI, body mass index; SD, standard deviation.

**Table 2 jfmk-09-00134-t002:** Descriptive data for the performance indicators.

Performance Indicators		Mean ± Standard Deviation (N = 15)	
Cross-sectional area (cm^2^)	Abductor hallucis (ABH)	3.16	0.65
	Adductor digiti minimi (ADD)	1.58	0.44
	Flexor digitorum brevis (FDB)	2.61	0.7
	Flexor hallucis brevis (FHB)	2.52	0.7
	Tibialis anterior (TA)	6.88	1.3
	Peroneal longus and brevis (PL)	4.94	1.14
	Posterior tibialis (PT)	5.32	1.03
	Extensor digitorum longus (EDL)	3.72	0.89
	Flexor hallucis longus (FHL)	2.75	0.9
	Flexor digitorum longus (FDL)	2.83	0.64
Muscle strength (N/kg)	Dorsiflexion	3.88	1.15
	Planter flexion	2.45	0.56
	Eversion	2.31	0.61
Toe grasp strength	% BW	47.4	14.4
Plantar superficial sensation (Filament size)	(1) Base of the 5th toe	3.42	0.66
	(2) Base of the 3rd toe	3.44	0.59
	(3) Base of the 1st toe	3.64	0.56
	(4) The 5th metatarsal joint	3.71	0.36
	(5) The 3rd metatarsal joint	3.65	0.44
	(6) The 1st metatarsal joint	3.58	0.47
	(7) Base side of the 5th metatarsal base	3.70	0.52
	(8) Base side of the navicular bone	3.42	0.50
	(9) Base in the center of the heel	4.17	0.66
Y-balance test (cm)	Reach distance (anterior)	65.3	4.2
	Reach distance (posteromedial)	112.4	6.5
	Reach distance (posterolateral)	110.4	6.3
	Composite	96.0	4.9

ABH, abductor hallucis; ADD, adductor digiti minimi; FDB, flexor digitorum brevis; FHB, flexor hallucis brevis; TA, tibialis anterior; PL, peroneal longus and brevis; PT, posterior tibialis; EDL, extensor digitorum longus; FHL, flexor hallucis longus; FDL, flexor digitorum longus.

**Table 3 jfmk-09-00134-t003:** Electromyography data during Y-balance test.

		Mean ± Standard Deviation (N = 15)	
Y-balance test (cm)	Reach distance (anterior)	65.3	4.2
Electromyography (% MVC)	ABH	50.2	20.3
	ADD	33.8	39.6
	TA	40.7	16.9
	PL	38.4	18.4
	PB	32.0	12.0
	MG	11.6	8.3
	SOL	52.8	33.5
	Reach distance (posteromedial)	112.4	6.5
	ABH	45.4	20.9
	ADD	26.1	25.3
	TA	45.3	15.0
	PL	41.3	16.2
	PB	33.3	10.9
	MG	9.9	7.6
	SOL	47.8	36.0
	Reach distance (posterolateral)	110.4	6.3
	ABH	31.8	13.3
	ADD	29.8	21.1
	TA	46.9	11.0
	PL	36.6	15.3
	PB	28.4	10.0
	MG	9.8	8.2
	SOL	52.8	32.6

ABH, abductor hallucis; ADD, adductor digiti minimi; TA, tibialis anterior; PL, peroneus longus; PB, peroneus brevis; MG, medial gastrocnemius; SL, soleus.

**Table 4 jfmk-09-00134-t004:** Correlations of lifesavers’ physical functions with reach distance during the Y-balance test (N = 15).

Y-Balance Test	Reach Distance	Anterior		Posteromedial		Posterolateral		Composite	
		r	*p*	r	*p*	r	*p*	r	*p*
Cross-sectional area (muscles)	Abductor hallucis (ABH)	−0.085	0.762	0.246	0.377	0.206	0.461	0.174	0.535
	Adductor digiti minimi (ADD)	−0.006	0.983	0.311	0.259	0.148	0.599	0.201	0.474
	Flexor digitorum brevis (FDB)	0.140	0.618	−0.006	0.984	−0.006	0.983	0.035	0.902
	Flexor hallucis brevis (FHB)	0.184	0.512	−0.033	0.907	−0.163	0.563	−0.032	0.909
	Tibialis anterior (TA)	0.064	0.821	−0.243	0.384	−0.302	0.274	−0.22	0.432
	Peroneal longus and brevis (PL)	0.068	0.81	−0.139	0.621	−0.359	0.188	−0.197	0.481
	Posterior tibialis (PT)	−0.032	0.908	−0.251	0.366	−0.351	0.2	−0.272	0.327
	Extensor digitorum longus (EDL)	0.089	0.752	−0.322	0.242	−0.455	0.089	−0.313	0.255
	Flexor hallucis longus (FHL)	0.06	0.831	0.244	0.382	0.133	0.636	0.183	0.514
	Flexor digitorum longus (FDL)	0.069	0.806	−0.021	0.94	−0.007	0.979	0.007	0.98
Muscle strength	Dorsiflexion	0.438	0.103	0.428	0.112	0.443	0.098	0.505	0.055
	Planter flexion	0.276	0.319	0.572 *	0.026	0.695 *	0.004	0.632 *	0.012
	Eversion	−0.051	0.857	0.097	0.731	−0.038	0.892	0.012	0.965
Toe grasp strength	% BW	0.501	0.057	0.579 *	0.024	0.588 *	0.021	0.652 *	0.008
Plantar superficial sensation	(1) Base of the 5th toe	0.083	0.769	−0.16	0.569	−0.193	0.49	−0.131	0.642
	(2) Base of the 3rd toe	−0.036	0.9	−0.108	0.702	−0.162	0.564	−0.128	0.65
	(3) Base of the 1st toe	−0.164	0.56	−0.375	0.168	−0.331	0.228	−0.356	0.193
	(4) The 5th metatarsal joint	−0.474	0.075	−0.607 *	0.016	−0.47	0.077	−0.607 *	0.016
	(5) The 3rd metatarsal joint	−0.522 *	0.046	−0.431	0.109	−0.262	0.346	−0.452	0.091
	(6) The 1st metatarsal joint	−0.409	0.131	−0.552 *	0.033	−0.511	0.051	−0.581 *	0.023
	(7) Base side of the 5th metatarsal base	−0.265	0.341	−0.219	0.433	−0.186	0.508	−0.252	0.365
	(8) Base side of the navicular bone	−0.18	0.521	−0.277	0.317	−0.285	0.303	−0.297	0.282
	(9) Base in the center of the heel	−0.297	0.282	−0.495	0.061	−0.449	0.093	−0.498	0.059
Electromyography	Abductor hallucis (ABH)	0.07	0.805	−0.224	0.422	−0.515 *	0.05		
	Adductor digiti minimi (ADD)	0.150	0.593	−0.145	0.607	−0.041	0.884		
	Tibialis anterior (TA)	0.143	0.611	0.257	0.355	0.253	0.363		
	Peroneal longus (PL)	−0.233	0.404	0.075	0.791	−0.065	0.817		
	Peroneal brevis (PB)	−0.214	0.444	0.117	0.678	0.127	0.652		
	Medial gastrocnemius (MG)	0.075	0.79	0.47	0.077	0.195	0.487		
	Soleus (SOL)	0.046	0.869	−0.116	0.681	−0.195	0.487		

* The *p*-value was significant at the 0.05 level.

## Data Availability

The data presented in this study are available on request from the corresponding author. The data are not publicly available due to Personal protection.

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
