# Peer review of "Relationship between Dynamic Balance and Physical Characteristics and Functions in Elite Lifesaving Athletes"

_jfmk, 2024, doi:10.3390/jfmk9030134_

Round 1

Reviewer 1 Report

Comments and Suggestions for Authors

Reviewer 2 Report

Comments and Suggestions for Authors

Dear,

please find my comments attached.

Kind regards

Reviewer 3 Report

Comments and Suggestions for Authors

The study investigates lifesavers' foot function and relates their performance with a dynamic balance test. It is an interesting paper with potential applications to other sports performed on sand or unstable ground. However, in my opinion, it has a considerable methodological weakness: there is no control/comparison group, such as healthy individuals or elite athletes from other sports that perform on stable ground. By now it is extremely descriptive.

Specific comments below.

The title could be improved to better reflect the aim of the paper. Are physical characteristics (title) and physical function (abstract) synonymous?

Considering the title, abstract, and introduction, different terms are used, and it is unclear what they refer to, even if they regard the same aspects, such as physical characteristics, physical function, and foot muscle morphology and function.

Dynamic balance is repeated in the title and keywords.

Lines 15-17 - What is measured is summarized, but dynamic balance is not enumerated.

Materials and Methods

Line 47 - What characterizes the sample as elite lifesavers? Besides, how many years of practice do they have? At the time they were assessed, what was the volume of training (sessions per week and number of hours per week)?

Lines 63-67 - Considering the importance of the gastrocnemius/soleus for ankle joint performance, it is unclear why their cross-sectional area was not determined as an extrinsic muscle.

Lines 76-86 - First, it is mentioned that external rotation ankle isometric strength was measured, then in line 86, it is referred to as abduction strength. The combination of both is called eversion. It is suggested to use the same terminology throughout the text.

Line 91 - It seems more adequate to refer to toes grasp force, considering that 5 toes are being measured.

It is unclear why the right foot was measured instead of the dominant leg side.

Figure 2 can be improved in size.

Confirm in the text whether the Y-balance test was applied on stable ground or on sand.

Line 130 - Suggest using the term "lower limb length" throughout the text to distinguish from actual leg length.

Lines 132-135 and subsection 2.7 contain partly duplicated information.

Results

Table 2 - Correct "planter." What are the units for muscle cross-sectional area? For plantar superficial sensation and the Y-balance test, present the units.

Table 3 - Present units.

Table 4 - Suggest presenting a correlation color map.

Reviewer 4 Report

Comments and Suggestions for Authors

Dear AuthorsThanks a lot for giving me the opportunity to revise your manuscrip, the following suggestion may improve the quality of your work.

Introduction

* While it introduces the concept of balance in lifesaving athletes, it could more precisely delineate the specific research gap this study aims to fill.

* The introduction would benefit from a clearer statement of the study's objectives and hypotheses. Explicitly defining these early on would enhance the reader's understanding of the study's direction.

*in the balance task, the relationship of the two ankles play a pivot role. Please discuss about this concept that it is present also post injuries in line with the "bilateral consequences of unilateral injury’ hypothesis" , in which unilateral injury could lead to bilateral consequences due to a general reorganization of the sensorimotor system, and this might be observed as an altered postural control during a single-leg stance on the “healthy“limb. Please take in to consideration these article: 

- Deodato M, Coan L, Buoite Stella A, Ajčević M, Martini M, Di Lenarda L, Ratti C, Accardo A, Murena L. Inertial sensors-based assessment to detect hallmarks of chronic ankle instability during single-leg standing: Is the healthy limb "healthy"? Clin Biomech (Bristol, Avon). 2023 Jul;107:106036. doi: 10.1016/j.clinbiomech.2023.106036. Epub 2023 Jun 29. PMID: 37406582.

Groters S, Groen BE, van Cingel R, Duysens J. Double-leg stance and dynamic balance in individuals with functional ankle instability. Gait Posture. 2013 Sep;38(4):968-73. doi: 10.1016/j.gaitpost.2013.05.005. Epub 2013 Jun 26. PMID: 23810093.

Method

The manuscript does not clearly justify the chosen sample size or discuss how it ensures sufficient power 

- why it is not present an healthy control group?

-why did you assess only the right foot?

- the inclusion/exclusion criteria should be better explain

Discussion

*a more comprehensive discussion of the study's limitations, including the small sample size, the absence of control group and the assess of only one leg, would provide a more balanced view.

 *The discussion would benefit from more detailed suggestions for future research

Round 2

Reviewer 2 Report

Comments and Suggestions for Authors

Dear,

please find my comments attached.

Kind regards

Reviewer 3 Report

Comments and Suggestions for Authors

In my opinion the authors successfully addressed my main concerns and made significant improvements to the paper.

Author Response

Thank you for your consideration.

Reviewer 4 Report

Comments and Suggestions for Authors

Dear authors,

thanks to give me the opportunity to revise againe your paper. Unfortunately, my suggestion were not take in consideratio point by point. I will be more than happy to revise again your work with my previous suggestion. 

Round 3

Reviewer 4 Report

Comments and Suggestions for Authors

I thank authors for the revision done.